# Spatial structure favors microbial coexistence except when slower mediator diffusion weakens interactions

Alexander Lobanov[1†], Samantha Dyckman[1†], Helen Kurkjian[1,2], Babak Momeni[1*]

[1]Biology Department, Boston College, Boston, United States; [2]Department of Aquatic Ecology, Swiss Federal Institute of Aquatic Science and Technology, Dübendorf, Switzerland

**Abstract** Microbes often exist in spatially structured environments and many of their interactions are mediated through diffusible metabolites. How does such a context affect microbial coexistence? To address this question, we use a model in which the spatial distributions of species and diffusible interaction mediators are explicitly included. We simulate the enrichment process, examining how microbial species spatially reorganize and how eventually a subset of them coexist. In our model, we find that slower motility of cells promotes coexistence by allowing species to co-localize with their facilitators and avoid their inhibitors. We additionally find that a spatially structured environment is more influential when species mostly facilitate each other, rather than when they are mostly competing. More coexistence is observed when species produce many mediators and consume some (not many or few) mediators, and when overall consumption and production rates are balanced. Interestingly, coexistence appears to be disfavored when mediators are diffusing slowly because that leads to weaker interaction strengths. Overall, our results offer new insights into how production, consumption, motility, and diffusion intersect to determine microbial coexistence in a spatially structured environment.

**\*For correspondence:**
momeni@bc.edu

[†]These authors contributed equally to this work

## Editor's evaluation

This important study uses computational simulations to explore when spatial structure can promote the coexistence between different microbial species and when not, ultimately helping to explain diversity in microbial communities. The evidence supporting the conclusions is convincing, based on extensive parameter sweeps. The conclusion that spatial structure only promotes coexistence under certain conditions is a testable hypothesis that is very interesting to microbial ecologists quite broadly.

## Introduction

Microbes are rarely found in isolation in nature. Instead, they are found coexisting with one another in complex networks of interactions (*Ma et al., 2020*). Given the differences among taxa and the competitive forces that act between them, a fundamental question in microbial community ecology is how this coexistence is maintained (*Chesson, 2000b*; *Solé and Bascompte, 2006*; *Widder et al., 2016*). And because many important industrial, environmental, and health-related processes rely on microbial communities to function (e.g. anaerobic granules, microbial mats, and gut microbiota, respectively), understanding the conditions that favor microbial coexistence is critical to sustaining these systems.

Spatial structure and organization may shape coexistence via numerous mechanisms (*Tilman, 1994*; *Durrett and Levin, 1994*; *Tilman and Kareiva, 1998*; *Durrett and Levin, 1998*; *Kerr et al., 2002*; *Brockhurst et al., 2006*; *Amarasekare, 2003*), often by modulating the interactions among individuals. For example, in a spatially structured environment where progeny is more likely to be in the vicinity of parents, intensified intrapopulation competition can give less competitive species a chance to survive (*Chesson, 2000a*). In other conditions, spatial isolation can allow organisms with conflicting abiotic needs to flourish in appropriate environments (*Kim et al., 2011*; *Satoh et al., 2007*). The interplay between dispersal and competition can also allow coexistence between species that are more competitive growers and species that are better at dispersing and colonizing (*Tilman, 1990*).

Spatial heterogeneity has been invoked as a mechanism for microbial coexistence since the pioneering work by *Gause, 1934*. And although general concepts of coexistence are expected to apply equally to microbes, microbial communities may be affected by spatial structure in unique ways because of the scale and multiplicity of microbial interactions. An important and ubiquitous example of this are microbial interactions that are mediated via diffusible metabolites—including resources and metabolic byproducts. Spatial structure can stabilize these interactions and support coexistence, for example, by allowing cheaters to be excluded from beneficial interactions (*Momeni et al., 2013b*; *Pande et al., 2016*), or by permitting facilitative chemical interactions while preventing the inhibitory effects of an interacting organism's physical presence (*Kim et al., 2008*). And while it is clear that the outcomes of interactions via diffusible mediators in structured environments may depend on mediator diffusion rates (*Kümmerli et al., 2014*; *Allison, 2005*) and the larger network of antagonistic and cooperative interactions (*Nadell et al., 2016*), how such factors translate into community-level consequences is not well understood.

Prior reports that address coexistence of metabolically interacting microbes in a spatially structured environment are scarce. In an implicit model, *Murrell and Law, 2003* have shown in a modified Lotka–Volterra model that when interspecific competition operates over shorter distances than intraspecific competition a spatially structured environment can lead to species coexistence by allowing for aggregation. And in recent work with explicit modeling of space, *Weiner et al., 2019* examined coexistence in territorial populations interacting through diffusible mediators and found that metabolic tradeoffs allow for the coexistence of more species than the number of nutrients.

Our model is distinct from previous work in that we allow overlap and dispersal of populations through the shared space. Our motivation is to capture situations in which microbes can disperse inside a matrix that defines the spatial structure. An example of this is the mucosal layer of the digestive or respiratory tract, in which stratification is possible, yet the distribution of different species populations can overlap. Another example is in yogurt or cheese, where spatial structure exists, but populations are not territorial. We modify a previously developed mediator-explicit model (*Niehaus et al., 2019*) to account for spatial structure and the dispersion of species in the same space. Here, we limit our study to one-dimensional (1D) spatial structure as a starting point. We examine in our model conditions under which coexistence is favored. We should emphasize that even though we choose our parameters within the range of typical values observed among microbial communities, the purpose of this work is not to recapture a specific community. Instead, by examining a range of values for parameters such as metabolite diffusion and species dispersal, we hope to gain a better understanding of how rates of these processes can affect species coexistence.

## Results
### A spatial mediator-explicit model of microbial communities

In our mediator-explicit model, species interact through metabolites that they produce and/or consume (*Figure 1A, B*; *Niehaus et al., 2019*). Each species can produce a subset of metabolites and consume a subset of metabolites. Each of the metabolites in the shared environment can in turn influence any of the species by increasing or decreasing their growth rate (i.e. facilitation or inhibition, respectively) compared to how each species grows in the absence of interactions (*Niehaus et al., 2019*). We also assume that different interaction mediators additively influence the overall growth rate of each species (see Model description in Methods).

We assume a 1D spatial structure which preserves the spatial context but allows the diffusion of metabolites and dispersal of species. Multiple metabolites and species can be present in a single

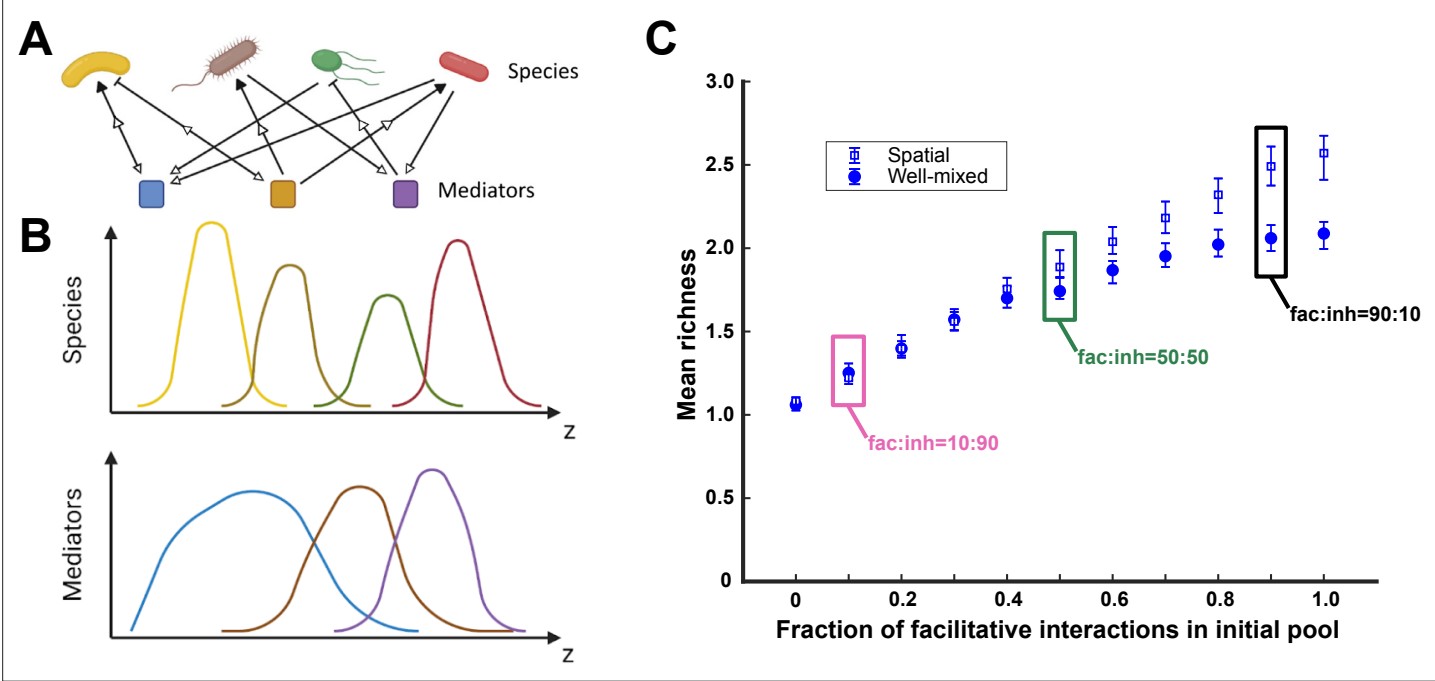

**Figure 1.** A spatial environment favors coexistence more when interspecies facilitation is prevalent in the initial species pool. (**A**) Species are engaged in metabolite-mediated interactions with other species. Each species produces a subset of mediators and consumes a subset. Each mediator can positively or negatively modulate the growth rate of the species it influences. In our model, consumption is present whenever there is an influence from a mediator on a species, regardless of whether the influence is facilitative or inhibitory. Production and consumption of mediators are indicated by open arrows. (**B**) In a one-dimensional (1D) spatial context, species and mediators are defined as functions of space that change over time because of population growth and dispersal as well as mediator production, consumption, and diffusion. A cartoon representation of the distributions of four species and three mediators are shown here over the spatial context (*z*). (**C**) Simulations were run at different ratios of facilitative to inhibitory interactions (fac:inh) for spatial (open blue squares) and well-mixed (filled blue circles) communities. Each ratio was run 500 times with the richness (number of species stably surviving at the end of a simulation) averaged over all the simulations. Each simulation started with 10 species and 5 mediators and ran for 100 generations. The error bars are 95% confidence intervals generated by bootstrapping 100 samples. Here, the species dispersal coefficient is $5 \times 10^{-9}$ cm²/hr. Boxes mark fac:inh ratios used in later simulations.

The online version of this article includes the following figure supplement(s) for figure 1:

**Figure supplement 1.** There is little change in community composition after 100 generations of growth.

**Figure supplement 2.** Shannon index shows the same overall trend as richness when comparing spatial versus well-mixed communities.

**Figure supplement 3.** Comparing the spatial distribution of species at different dispersal rates illustrates the impact of dispersal on coexistence.

**Figure supplement 4.** Species interactions and dynamics are different in spatial versus well-mixed environments, leading to different coexistence outcomes.

**Figure supplement 5.** Species interactions and dynamics are different in spatial versus well-mixed environments, even when inhibition is prevalent.

**Figure supplement 6.** Spatial distance between species can modulate the strength of their interaction.

**Figure supplement 7.** Within the same order of magnitude, the community's spatial extent does not have a large impact on spatial coexistence.

**Figure supplement 8.** Imposing a local carrying capacity favors species coexistence.

**Figure supplement 9.** Rearranging the order of species can modulate the strength of interspecies interactions and impact spatial coexistence outcomes.

location. Both metabolite diffusion and species dispersal are modeled as random walk processes, characterized with a diffusion coefficient and a dispersal coefficient, respectively. In a typical simulation, we start from an initial distribution in which populations occupy adjacent, non-overlapping spatial locations at low initial density. This choice is made to impose a reproducible initial condition that emphasizes the role of space. Each simulation starts with a network of interactions in which interaction strengths, production and consumption links, and production and consumption rates are assigned randomly. The initial pool typically contains 10 species and 5 interaction mediators. We simulate community enrichment through rounds of growth and dilution (*Niehaus et al., 2019*;

*Goldford et al., 2018*) for 100 generations, and assess the richness of each resulting community (i.e., the number of species stably persisting in the community). We have chosen 100 generations of growth, because we have observed that often this is enough to reliably decide which species stably persist in the community (*Figure 1—figure supplement 1*). At each dilution step, we assume that the overall spatial distribution of the community is preserved and all populations at all locations are diluted with the same factor. We recognize that this assumption is not universally true; however, we adopt it as an approximation, in the absence of additional information about a particular community. Such a dilution preserves some of the spatial structure of the community in the next round of growth and could represent a biofilm getting partially washed away by rain or in a microfluidic device, gut microbiota after a defecation event, or a broken-off portion of a granule initiating a new granule. We use a well-mixed version (*Niehaus et al., 2019*)—devoid of any spatial context—with the same set of parameters for species properties and interactions (i.e. consumption and production rates, basal growth rates, mediator influences, etc.) for all comparisons. *Figure 1—figure supplement 2* shows an example of the population distributions and dynamics during the course of enrichment. In a simple example, we show that interactions and subsequently the population dynamics are affected by growing in a well-mixed versus spatial environment (*Figure 1—figure supplement 3*). We explored the impact of the overall spatial extent of the community and found that within an order of magnitude of change, the outcomes remained the same (*Figure 1—figure supplement 4*).

The shift from interspecies competition to intraspecies competition can favor coexistence in a spatially structured environment. To assess this impact, we imposed a cap on total cell number at each location in space. As this cap became more restrictive, it suppressed the most competitive species and led to higher coexistence (*Figure 1—figure supplement 5*). Since our focus in this manuscript is the impact of interspecies interactions, in the rest of this manuscript we pick the total cell number cap at a level ($k_Y = 10^9$ cells/ml) that minimizes the impact of imposed intrapopulation competition.

## A spatial environment favors coexistence more when facilitation among species is prevalent

We first examined how the prevalence of facilitative versus inhibitory interactions impacted coexistence in spatial communities. In our simulations, we dictated the ratio of facilitative and inhibitory interactions in the initial pool of species. Our results show that, similar to a well-mixed environment, more facilitative interactions lead to higher richness in communities that emerge from enrichment (*Figure 1C*, along the *x*-axis). Additionally, we observe that spatial communities show more coexistence than well-mixed communities when facilitation among species is prevalent (*Figure 1C*, spatial versus well-mixed). The same pattern, although less pronounced, was present when instead of richness we used the Shannon index to assess the diversity of resulting communities (*Figure 1—figure supplement 2*). Our explanation is that species locally grow better when adjacent to a facilitative partner and grow worse when in the vicinity of an inhibitory partner. The resulting spatial self-organization in effect amplifies facilitative interactions and dampens inhibitory interactions, leading to more coexistence. This is supported by our data which shows that the position of specific species with respect to other species that facilitate or inhibit it can impact the population dynamics (*Figure 1—figure supplement 8*). Because of the marked impact of the fac:inh ratio (i.e. the ratio of the number of facilitative interactions to the number of inhibitory interactions), moving forward, we will examine three conditions, with equal fractions of facilitative and inhibitory influences (fac:inh = 50:50), mostly inhibitory (fac:inh = 10:90), or mostly facilitative (fac:inh = 90:10) to scope the impact on coexistence.

At low species dispersal, self-organization is one of the mechanisms that can lead to a difference between spatial and well-mixed communities (*Figure 1—figure supplement 3*). In a simplified interpretation, self-organization can be in the form of co-localization driven by facilitation or segregation driven by inhibition. In our simulations, we observed that co-localization had a stronger effect on coexistence. The positive influence was reinforced by more growth in the vicinity of the partner, leading to a stronger representation of facilitation in spatial communities. In contrast, segregation only had a modest effect on weakening the impact of inhibition. As a result, there is more similarity between well-mixed and spatial communities in the absence of strong facilitative interactions (*Figure 1C*).

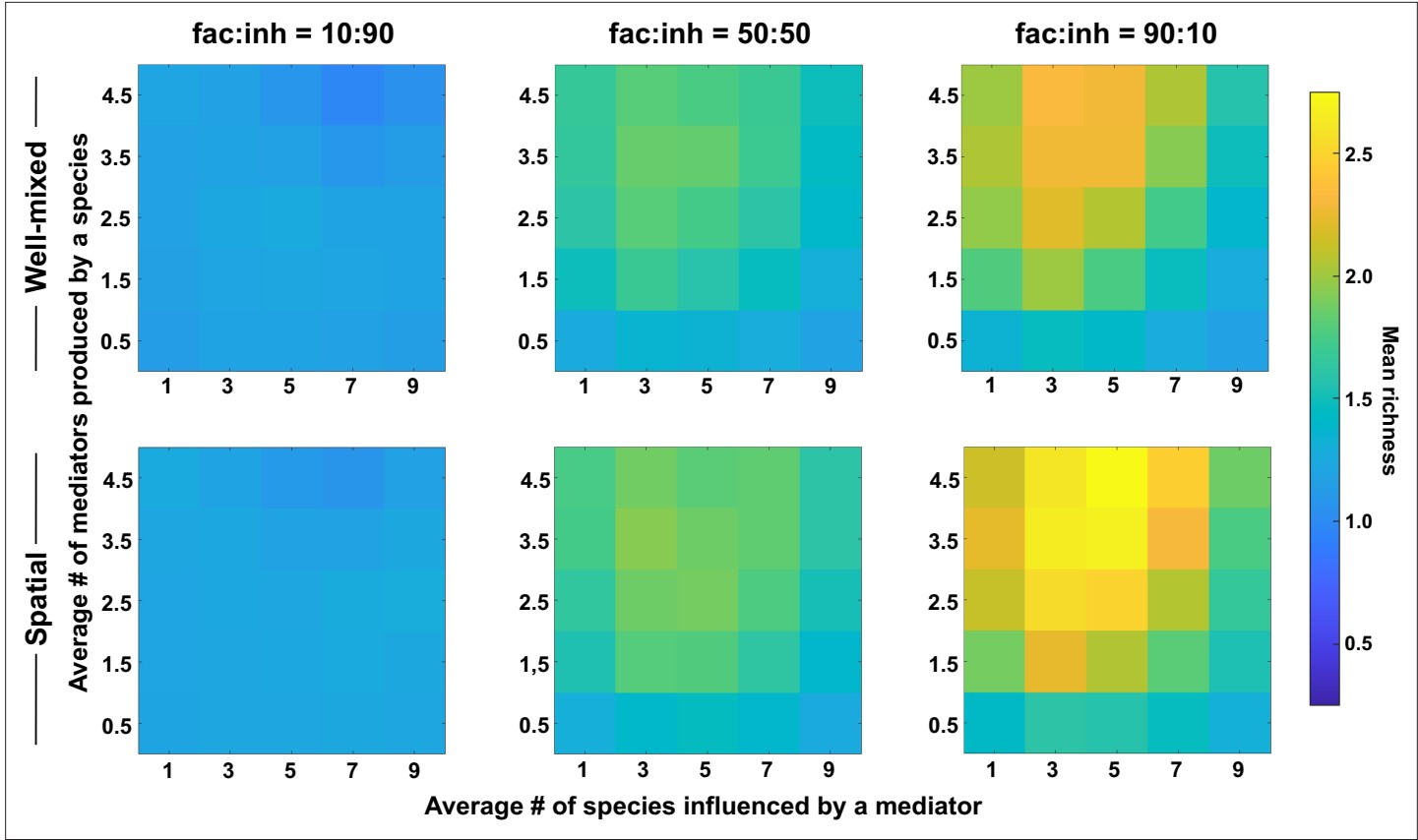

**Figure 2.** The number of metabolites produced and number of species influenced affect coexistence in spatial and well-mixed communities. Different ranges of production and mediator influence values were analyzed for both well-mixed and spatial communities at three different fractions of fac:inh influences in the initial pool of species (10:90, 50:50, and 90:10). Mean richness (i.e. average number of species stably present at the end of a simulation) was calculated for 500 simulated instances and marked on the color bar. Each simulation started with 10 species and 5 mediators and ran for 100 generations. The *x*-axis represents the average number of species influenced by a mediator and the *y*-axis represents the average number of mediators produced by each species. Other simulation parameters are listed in *Table 1*.

The online version of this article includes the following figure supplement(s) for figure 2:

**Figure supplement 1.** The number of metabolites produced and number of species influenced affect coexistence in spatial and well-mixed communities.

**Figure supplement 2.** Coexistence is favored when many metabolites are produced and influence an intermediate number of species, even with weaker interactions.

**Figure supplement 3.** Coexistence is favored when many metabolites are produced and influence an intermediate number of species, even with stronger interactions.

**Figure supplement 4.** Self-facilitation is prominent among low-diversity outcomes.

## Coexistence is favored when many metabolites are produced and influence an intermediate number of species

Because metabolites are at the center of interspecies interactions in our model, we examined their impact on spatial coexistence of the average number of metabolites produced by each species and the average number of species influenced by each mediator. We found that coexistence is favored when the number of metabolites produced is larger (*Figure 2*, along the *y*-axis). This effect was stronger when the metabolite influences were mostly facilitative (fac:inh = 90:10, versus 50:50 or 10:90). In contrast, coexistence achieved its maximum values at intermediate ranges of mediator influence (*Figure 2*, *x*-axis), that is lower coexistence was observed when each mediator influenced too many or too few species on average. We note that these trends were largely the same between spatial and well-mixed communities.

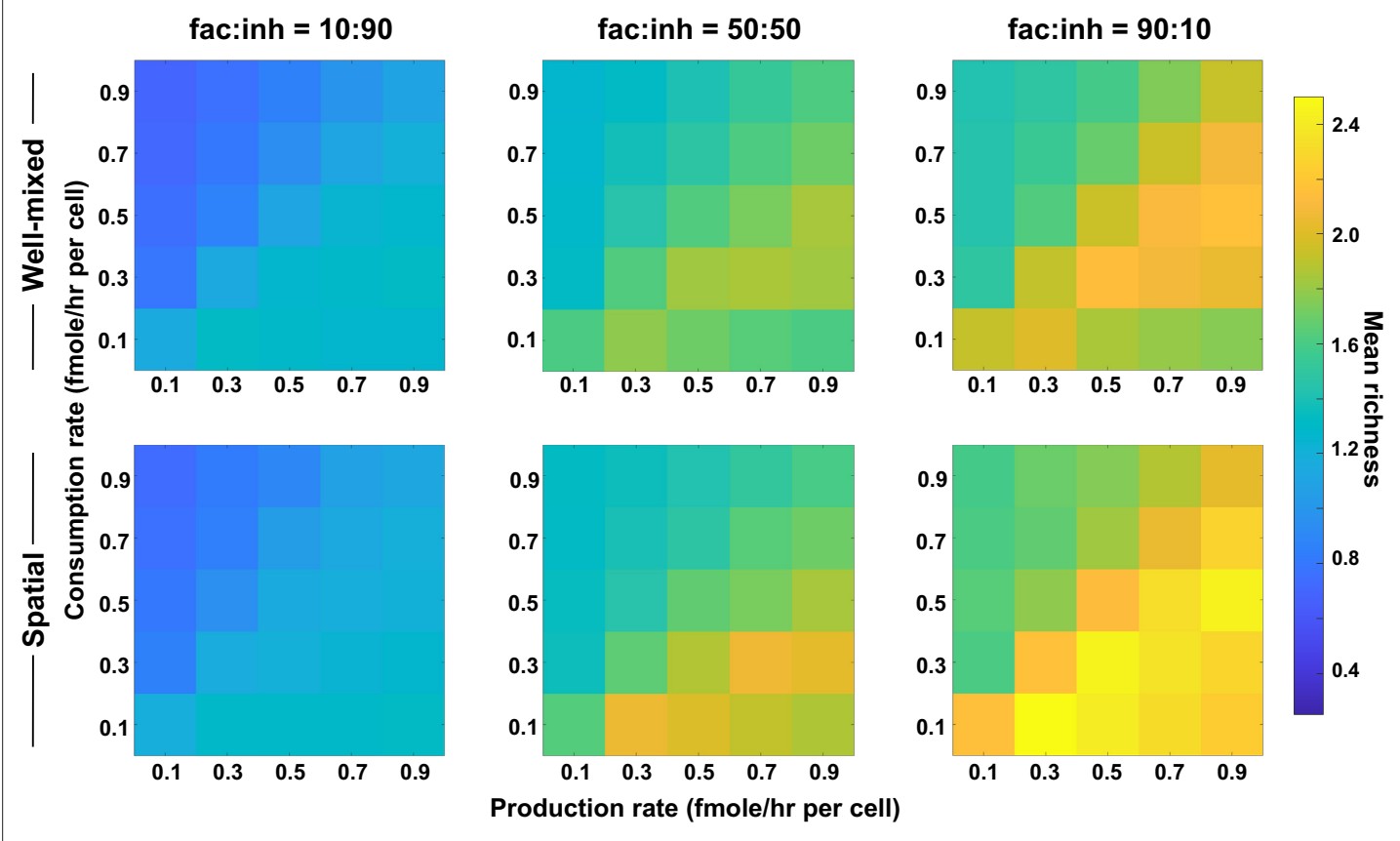

**Figure 3.** Coexistence is higher where there is a balance between production and consumption of mediators. Different average production and consumption rates were analyzed for both well-mixed and spatial communities at three different fractions of fac:inh influences in the initial pool of species (10:90, 50:50, and 90:10). Mean richness (i.e. average number of species stably present at the end of a simulation) is calculated for 500 simulated instances. Each simulation started with 10 species and 5 mediators and ran for 100 generations. Color bar represents the average richness. The *x*-axis represents the average production rate of mediators and the *y*-axis represents the average consumption rate of mediators. Other simulation parameters are listed in *Table 1*.

The online version of this article includes the following figure supplement(s) for figure 3:

**Figure supplement 1.** In spatial communities with many facilitation interactions among species, coexistence is favored at lower consumption rates of mediators.

**Figure supplement 2.** Coexistence is higher where there is a balance between production and consumption of mediators independent of the interaction strength (here, with weaker interactions).

**Figure supplement 3.** Coexistence is higher where there is a balance between production and consumption of mediators, independent of the interaction strength (here, with stronger interactions).

Our explanation is that a larger range of production offers more opportunities for interaction, which through the enrichment process lead to the selection of facilitative subsets that coexist (*Niehaus et al., 2019*). A low mediator influence range works in the opposite direction, reduces opportunities for interactions and results in lower coexistence. Very high mediator influence range potentially leads to more self-facilitation (i.e. producing a metabolite that is beneficial to the producer species), which our data suggest can lead to take-over by a single species and a lower coexistence as a result (*Figure 2—figure supplement 1*).

## Coexistence is higher when there is balance between production and consumption of mediators

We next asked how the rates of production and consumption of mediators would influence coexistence. To address this question, we surveyed a range of average rates of production and consumption. We observed that the highest levels of coexistence occurred when there was a balance between

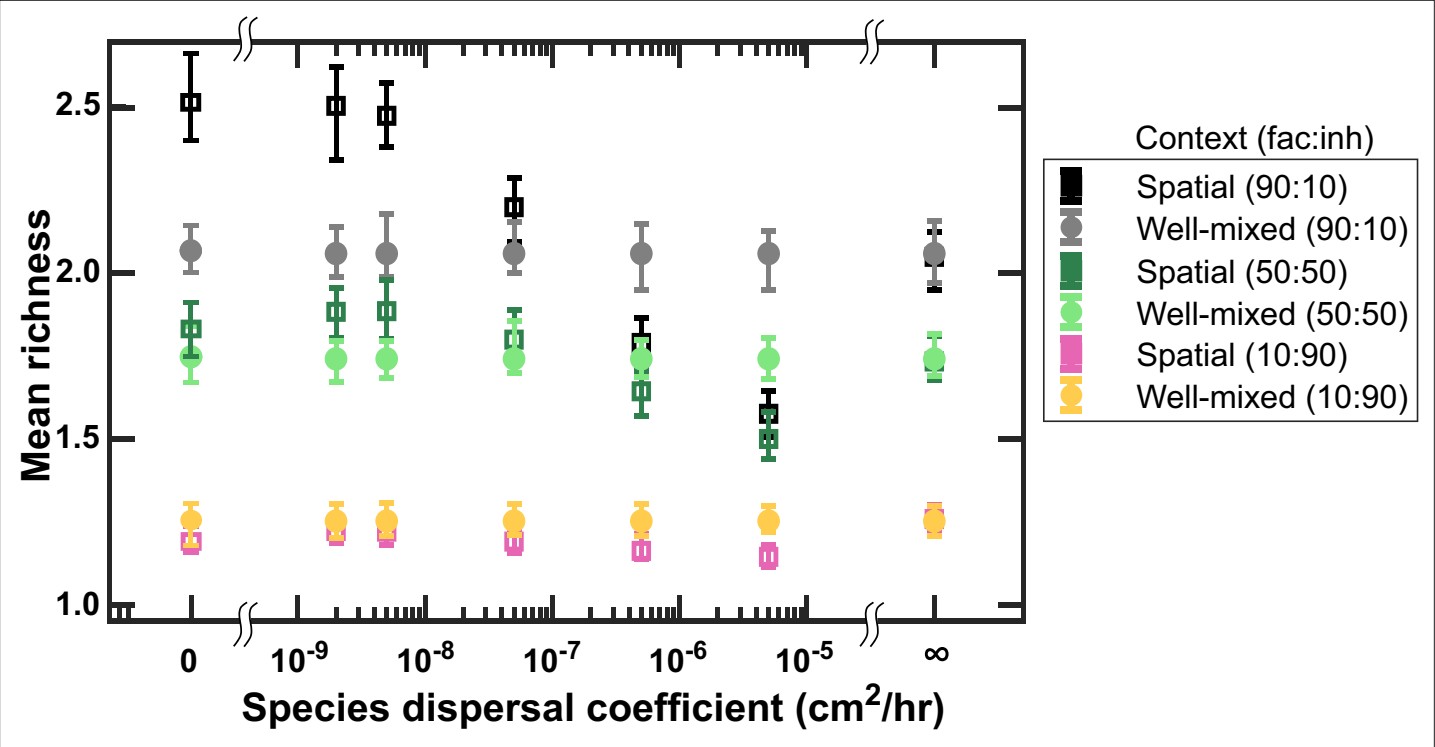

**Figure 4.** Lower dispersal rates allow more microbial coexistence. Communities in spatially structured environments were simulated with different dispersal coefficients at three different fractions of fac:inh influences (10:90, 50:50, and 90:10). Mean richness (i.e. average number of species stably present at the end of a simulation) was calculated for 500 simulated instances. Each simulation started with 10 species and 5 mediators and ran for 100 generations. Other simulation parameters are listed in **Table 1**. The error bars are 95% confidence intervals generated by bootstrapping 100 samples.

The online version of this article includes the following figure supplement(s) for figure 4:

**Figure supplement 1.** Lower dispersal rates allow more microbial coexistence.

**Figure supplement 2.** Self-facilitation contributes to lower richness at intermediate levels of dispersal.

**Figure supplement 3.** Rearranging the order of species affects spatial coexistence.

**Figure supplement 4.** Rearranging the order of species affects spatial coexistence more in communities in which facilitation is prevalent.

consumption and production rates among species, with slightly higher production than consumption (**Figure 3**).

Our justification for the observed pattern is that in one extreme where production is too high (lower right corner of each plot), mediators will build up in the environment. This will put the community in a regime in which consumption is not enough to create a feedback, that is 'reusable mediators' as discussed in **Niehaus et al., 2019**, which leads to lower coexistence. In the other extreme, when consumption is too high (upper left corner of each plot), metabolites that mediate the interactions will be depleted from the environment, leading to an effectively weaker interaction and thus lower coexistence. However, when production is slightly higher than consumption, metabolite quantities are sufficient to create strong interactions and facilitation feedbacks, leading to higher coexistence. While coexistence is slightly higher in the spatial communities compared to well-mixed ones, the production–consumption trends apply equally to spatial and well-mixed communities, as expected.

### Limited species dispersal in a spatial environment allows more coexistence, especially when facilitation is common

Because species dispersal is a major factor in preserving community spatial structure, we examined how the dispersal coefficient affected coexistence outcomes. For this, we kept the diffusion coefficient of the mediators fixed and surveyed mean richness among many instances of communities randomly assembled (*n* = 500). When the diffusion coefficient for species approaches zero and cells remain in their original spatial location, we observe higher levels of coexistence (**Figure 4**). We also

observed that the impact of lower dispersal is stronger in communities in which most interactions are facilitative rather than inhibitory. Our explanation is that lower dispersal rates mean that species grow best in spatial locations that are more supportive for their growth, which is in the vicinity of their beneficial partners and away from competitors or inhibitors. As discussed in *Figure 1*, such self-organization effectively amplifies facilitative interactions and de-emphasizes inhibitory interactions, leading to a higher coexistence. This is also consistent with the observation that the effect of dispersal rate is strongest when the proportion of facilitative interactions is highest. As the dispersal coefficient increases, the self-organization gets washed away by dispersal of cells to less than ideal locations for their growth and its benefit for coexistence diminishes.

At intermediate levels of dispersal, the trend reversed and well-mixed communities showed more coexistence compared to spatial communities. This is interesting because at the limit of extremely rapid diffusion (shown with a '∞' sign in *Figure 4*) when we kept the species distribution uniform across the spatial extent, coexistence outcomes matched the well-mixed case, as expected. We found two factors that contributed to this trend. The first contribution came from longer-term changes in dynamics at intermediate levels of dispersal. Even after 100 generations, which is the typical extent of our studies, at intermediate levels of dispersal (e.g. $D_{Cell} = 5 \times 10^{-6}$ cm$^2$/hr), the spatial distribution of populations is still changing considerably. As a result, our strict criteria for stable coexistence removes some of the populations that are still temporally not stable enough, leading to a lower overall assessment of coexistence in these cases. To show this, we examined the range of dispersal coefficients again, but kept all the species that were present after 100 generations, rather than those with stable population fractions at that point (see Model implementation in Methods). The results show that higher dispersal coefficients using this measure lowers the richness of resulting communities (based on presence, rather than stable presence), but not below the levels expected from well-mixed communities (*Figure 4—figure supplement 1*). As a second factor, we hypothesized

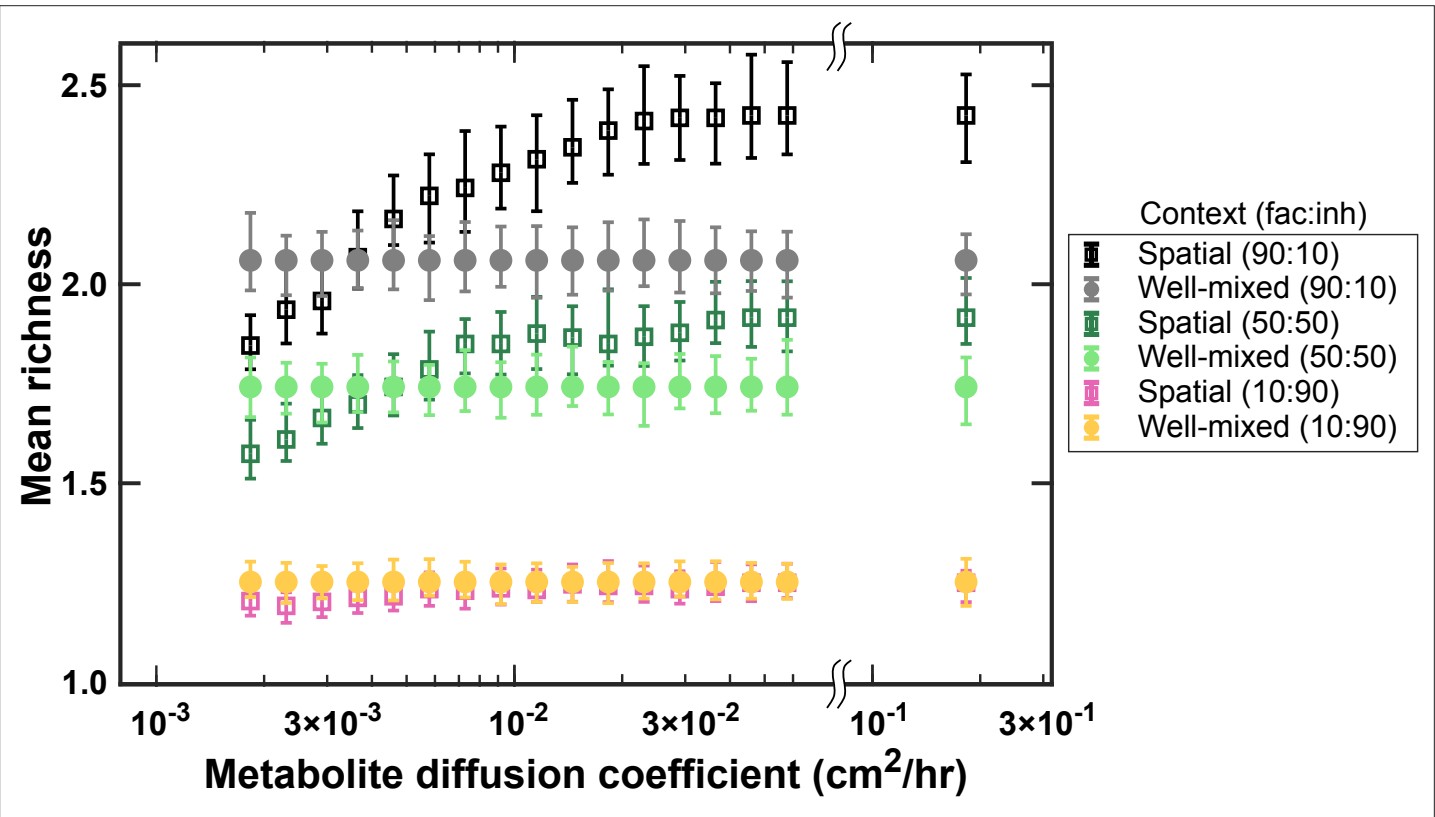

**Figure 5.** At higher diffusion coefficients of mediators more coexistence is possible. A range of metabolite diffusion coefficients were simulated in spatial communities (squares) at three different fractions of fac:inh influences (10:90, 50:50, and 90:10). We simulated corresponding well-mixed communities (circles) for comparison. Each condition was run 500 times with the richness (number of species stably surviving at the end of a simulation) averaged over all the simulations. Each simulation started with 10 species and 5 mediators and ran for 100 generations. Other simulation parameters are listed in *Table 1*. The error bars are 95% confidence intervals generated by bootstrapping 100 samples.

that self-facilitation interactions contribute to the decrease in coexistence at intermediate dispersal levels. Our rationale was that self-facilitation interactions are amplified in communities in which the spatial context is preserved, because the distribution of producers matches the distribution of self, but not other recipients in such a case. This can lead to community overtake by a self-facilitating species. This effect will be weaker in communities at intermediate dispersal rates: at low dispersal rates self-facilitating species will be more confined in space and some of the metabolite will leak out to other species; in the other extreme, in the very high dispersal rates all distributions will become uniform and the distinction between self and others diminishes. To test this, we tested weaker self-facilitation links in our simulations and observed that this change led to higher coexistence in communities with intermediate dispersal coefficients but not in well-mixed communities or communities with low dispersal coefficients (*Figure 4—figure supplement 2*). It is a matter of debate how prevalent self-facilitation interactions are within microbial communities. Self-facilitation interactions do exist, for example when a species breaks down a recalcitrant substrate such as cellulose into smaller molecules that can be beneficial. However, if they are not as prevalent as what our model assumes, some of our predictions might be affected.

## Coexistence is disrupted when the diffusion of mediators is too slow

The rate of diffusion of metabolites also has the potential to affect coexistence. We investigated coexistence over a range of mediator diffusion coefficients. We still typically observe a higher mean richness for spatial communities compared with the well-mixed communities (*Figure 5*). However, unlike the conventional wisdom, as the diffusion of mediators becomes slower, coexistence in spatial communities decreases. At low diffusion coefficients, coexistence drops even below that of corresponding well-mixed communities. We associate this trend to weaker effective interactions among species at lower diffusion coefficients. Mediators that are involved in facilitation play a major part in allowing coexistence of species (*Niehaus et al., 2019*); if these mediators get consumed by nearby species and do not travel long enough to reach other members of the community, the interaction-driven mechanism of coexistence is disrupted.

## Discussion

Our results dispel the common presumption that a spatially structured environment will universally lead to more coexistence. We find that, compared to a well-mixed environment, a spatial environment can favor or disfavor coexistence depending on the balance between species dispersal and the diffusion of interaction mediators. Interestingly, a lower species dispersal rate favors coexistence, but this effect can be diminished or even reversed if accompanied by low mediator diffusion rates. Coexistence is favored when species have a broad range of consumption and an intermediate range of production of interaction mediators. Additionally, we predict more coexistence when there is a balance between overall production and consumption rates for mediators.

The spatial structure of microbial communities has been extensively studied for example in simulating the development of biofilms (*Wang and Zhang, 2010*; *Kreft et al., 2001*; *Xavier and Foster, 2007*; *von der Schulenburg et al., 2009*), for specific interactions among species (*Momeni et al., 2013b*; *Momeni et al., 2013a*; *Kang et al., 2014*), or for modeling game-theory dynamics (*Nakamaru et al., 1997*; *Brauchli et al., 1999*; *Saxer et al., 2009*; *Hauert and Doebeli, 2004*). However, as there is often a tradeoff between the incorporation of detailed mechanisms and generality of conclusions (*Levins, 1966*; *Momeni et al., 2011*), we chose in this work to explore a simple, general model of chemically mediated microbial interactions. We assumed, for example, that mediators affected species by additively influencing their growth rates. Although it is possible (and even probable) that mediator effects could be multiplicative, nonlinear, or otherwise context dependent and that they may impact other model parameters, we chose here to present what we felt to be the simplest case. Exploration of alternative implementations of mediator effects would make a fascinating follow-up to this work.

We have made assumptions in our model to simplify the configuration and make the analyses and interpretations easier. We asked if making these assumptions more realistic would affect our conclusions. For example, we have assumed no carrying capacity limit for the growth of our populations. We explored the effect of imposing a total population limit, enforced at each spatial location, and found

that it did alter our conclusions (*Figure 1—figure supplement 5*). However, because the relationship between carrying capacity and coexistence has been explored extensively elsewhere, we chose parameters to minimize this impact, allowing us to focus on other interspecies interactions (beyond competition) and relative rates of diffusion and dispersal. We also tested the impact of the spatial extent of the community ($Z$), and observed that our results were largely unaffected if the community's spatial extent was changed by an order of magnitude (*Figure 1—figure supplement 4*). The effect of larger changes in the spatial extent can be examined by scaling the diffusion and dispersal coefficients accordingly.

Beyond the details of our assumptions, there are also alternative representations of interactions among species, including a simplified Lotka–Volterra model and its variations (*Wangersky, 1978*), a consumer-resource model (*Goldford et al., 2018*; *Marsland et al., 2019*), or a reduced metabolic model (*Liao et al., 2020*). There are trade-offs in tractability and complexity in choosing which model to use. Our reasoning for adopting the mediator-explicit model was to (1) explicitly include metabolites that mediate the interactions in the model (*Momeni et al., 2017*); (2) incorporate both metabolites that support the growth of other species as well as those that are inhibitory, such as waste products and toxins (*Momeni et al., 2017*); and (3) keep the model simple to allow a clear interpretation of mechanisms and processes when analyzing the results. We think it will be worthwhile to compare the predictions of other models to clarify what assumptions are necessary to generate the trends we have obtained and how general the conclusions are.

If spatial organization of cells matters, we also expect that the initial spatial position of species in the community impacts coexistence. To test this, we started from 100 simulations instances and in each case, we tried 100 rearrangements, each obtained by shuffling the spatial position of species, while keeping the species properties and interactions intact. Interestingly, in many cases coexistence was affected (*Figure 4—figure supplement 3*), indicating that the adjacency to partners is an important determinant of spatial coexistence (as also suggested by *Figure 1—figure supplement 3* and *Figure 1—figure supplement 7*). When we examined the effect of the fac:inh ratio on these outcomes, we observed that larger changes in richness when facilitation interactions were more prevalent in the community (*Figure 4—figure supplement 4*), which aligns with many of our other results showing that facilitation amplifies the positive effect of spatial structure on coexistence. Although these results are tantalizing, a detailed examination of the spatial organization of populations and metabolites within the community requires a dedicated investigation and is beyond the scope of this work.

Finally, our model assumes that dispersal and diffusion rates are uniform across species and metabolites, respectively. However, dispersal ability can vary widely across microbial taxa, depending on cell size, motility type, chemotaxis, quorum sensing, and other factors. And how the dispersal rates of individuals scale up to affect population- and community-level dynamics is not well understood. Likewise, the diffusion rates of metabolites have the potential to vary greatly with molecule size and shape. Although outside the scope of this work, we are exploring heterogeneity in these rates of movement as an interesting follow-up.

Overall, we believe this work revisits how spatial structure—and spatial self-organization—affects community assembly and coexistence. In our model, which emphasizes the contributions of interspecies interactions, we find that the impact of spatial structure on coexistence largely arises from two processes: (1) spatial self-organization, which can improve coexistence by favoring facilitation over inhibition, and (2) localization of interactions, which can promote coexistence in association with self-organization or hamper coexistence by slowing down and weakening species interactions.

## Methods
### Model description

Our model is an extension of a model introduced earlier (*Niehaus et al., 2019*) in which a set of species interact with each other through diffusible mediators. Each mediator is produced by a subset of species, consumed by a subset of species, and has a positive or negative influence on the growth rate of some species (*Figure 1*).

$$\frac{dS_i\,(z,t)}{dt} = D_{Cell}\frac{d^2S_i\,(z,t)}{dz^2}$$

$$+ \left(1 - \frac{\sum_{j=1}^{M} S_j\,(z,t)}{k_Y}\right)\left[r_{i0} + \sum_{j=1}^{M} \rho_{ij}\left(\frac{C_j(z,t)}{k_{sat}}\delta_{ij}^- + \frac{C_j(z,t)}{C_j(z,t)+k_{sat}}\delta_{ij}^+\right)\right]S_i(z,t)$$

$$\frac{dC_j\,(z,t)}{dt} = D_{Med}\frac{d^2C_j\,(z,t)}{dz^2} + \sum_{j=1}^{M}\left[\beta_{ji}S_i\,(z,t) - \alpha_{ji}S_i\,(z,t)\right]$$

Here, $S_i$ is the spatiotemporal density of species $i$ ($i = 1, …, N_c$). $C_j$ is the spatiotemporal concentration of mediator $j$ ($j = 1, …, N_m$). $D_{Cell}$ is the dispersal coefficient for cells. $D_{Med}$ is the diffusion coefficient for mediators. $k_Y$ is the local carrying capacity (for the total density of cells) at each location. $\rho_{ij}$ is the interaction coefficient expressed as the impact of mediator $j$ on species $i$. Additionally,

$$\delta_{ij}^- = \begin{cases} 1, \rho_{ij} < 0 \\ 0, \rho_{ij} > 0 \end{cases} \quad \text{and} \quad \delta_{ij}^+ = \begin{cases} 0, \rho_{ij} < 0 \\ 1, \rho_{ij} > 0 \end{cases}$$

$k_{sat}$ is the interaction strength saturation level. $\beta_{ji}$ and $\alpha_{ji}$ are average production rate and consumption rates, respectively, between species $i$ and mediator $j$. Similar to **Niehaus et al., 2019**, each species

**Table 1.** Parameters used in our simulations are listed.

| Parameter | Value (unit) |
|---|---|
| Number of instances examined ($N_s$) | 500 |
| Number of cell types in the initial pool ($N_c$) | 10 |
| Number of interaction mediators ($N_m$) | 5 |
| Total initial cell density ($TID$) | $10^4$ (cells/ml) |
| Interaction strength saturation level ($k_{sat}$) | $10^4$ (cells/ml) |
| Population extinction threshold ($ExtTh$) | 0.1 (cells/ml) |
| Population dilution threshold ($DilTh$) | $10^7$ (cells/ml) |
| Consumption rate ($\alpha_{ij}$) | 0.075–2.25 (fmol per cell per hour; avg. 0.15) Stochastic with a uniform distribution |
| Production rate ($\beta_{ij}$) | 0.1–0.2 (fmol per cell per hour; avg. 0.1) Stochastic with a uniform distribution |
| Probability of production link per population ($q_p$) | 0.5 |
| Probability of influence link per mediator ($q_c$) | 0.5 |
| Maximum interaction strength magnitude ($r_{int,0}$) | 0.2 (1/hr) |
| Basal growth rate of species ($r_0$) | 0.1–0.2 (1/hr); stochastic with a uniform distribution |
| Number of generations for enrichment ($nGen$) | 100 |
| Dispersal coefficient for cells ($D_{Cell}$) | $5 \times 10^{-9}$ (cm²/hr) |
| Diffusion coefficient for mediators ($D_{Med}$) | $1.8 \times 10^{-2}$ (cm²/hr) |
| Local carrying capacity per dz ($k_Y$) | $10^9$ cells/ml |
| Total community spatial extent ($Z$) | 0.5 cm |
| Spatial resolution for species distributions ($dz$) | 0.005 cm |
| Cell growth update and uptake timescale ($dtau$) | 0.01 hr |
| Mediator diffusion time-step ($dt$) | $0.1 dz^2/D_{Med}$ |
| Cell dispersal simulation time-step ($dc$) | $0.1 dz^2/D_{Cell}$ |

has a basal growth rate (in the absence of interactions with other species), and influential mediators additively modulate this growth rate. At $z = 0$ and $z = Z$, no-flow boundary conditions are enforced for both species and mediators by setting the local spatial derivatives of these parameters to zero at the boundaries.

Typical parameters used in our simulations (unless otherwise stated) are listed in *Table 1*. These parameters are chosen in the expected realistic range of values; for example, the typical diffusion coefficient of small molecules in water is in the range of 100–1000 µm²/s, and we have used 500 µm²/s as a generic value. When comparing spatial communities with their well-mixed counterparts, exactly the same parameters for basal growth rates, production and consumption rates, mediator influences, and networks of production and consumption are used. This choice is made to reduce the stochasticity caused by other parameters and to focus only on the impact of spatial structure.

To sample different possibilities, the interaction terms as well as production and consumption rates are randomly assigned in each instance of the simulation. Similar to our previous work (*Niehaus et al., 2019*), the production/consumption matrices are random, that is each element of the matrix has a binomial distribution with a fixed probability of being present ($q_p$ and $q_c$ for production and consumption/influence links, respectively). The production and consumption rates have a uniform distribution between 0.5 and 1.5 times a set value each ($\beta_{ij}$ and $\alpha_{ij}$ for production and consumption rates, respectively). The interaction matrix which represents the influence of mediators on species has the same structure as the consumption matrix. The magnitude of the influence in this matrix has a uniform distribution between 0 and a maximum value, $r_{int,0}$ . The sign of the influence is chosen from a binomial distribution based on the ratio of fac:inh.

## Model implementation

We solve the equations in 'Model description' numerically in Matlab using a finite difference discrete version of the equations. Mediator diffusion and cell dispersal take place often at very different time scales. To simulate these processes, we use different numerical time-steps to update the mediator and cell distributions. To allow flexibility in modeling different diffusion and dispersal coefficients, we used asynchronous updates with two independent time-steps: one for updating the diffusion of metabolites and another one for growth and dispersal of cells. The source codes are shared for transparency and reproducibility (see Code availability).

To assess coexistence, we use a criterion similar to *Niehaus et al., 2019*. In short, any species whose density drops below a pre-specified extinction threshold (*ExtTh*) is considered extinct. Among species that persist throughout the simulation, only those are considered to coexist whose relative frequency does not decrease by more than 10% of its value in the last 20 generations of the simulation. We consider these species to be 'stably present' in the community. Species whose relative frequency declines faster than this threshold are assumed to go extinct later and are not considered to be part of coexisting communities. The only exception to this criterion is the data in *Figure 4—figure supplement 1*, in which all present species (rather than stably present species) are included in the assessment of final richness.

## Statistics

Mean richness values are calculated by averaging the richness values calculated over all simulated instances for a given condition. Confidence intervals for mean richness values are calculated by bootstrapping over all simulated instances for a given condition. The standard routine in Matlab, bootci, is used in all cases for bootstrapping.

## Acknowledgements

BM would like to acknowledge the Award for Excellence from Smith Family Foundation that supported this work. BM would like to also thank the biology department at Boston College for their continued support. We would like to acknowledge the support for AL through an Undergraduate Research Fellowship from Boston College.

## Additional information

### Competing interests
Babak Momeni: Reviewing editor, eLife. The other authors declare that no competing interests exist.

### Funding

| Funder | Grant reference number | Author |
|---|---|---|
| Richard and Susan Smith Family Foundation | | Samantha Dyckman Helen Kurkjian Babak Momeni |
| Boston College | URF | Alexander Lobanov |

The funders had no role in study design, data collection, and interpretation, or the decision to submit the work for publication.

### Author contributions
Alexander Lobanov, Conceptualization, Software, Formal analysis, Validation, Investigation, Visualization, Methodology, Writing – original draft, Writing – review and editing; Samantha Dyckman, Conceptualization, Software, Formal analysis, Investigation, Visualization, Methodology, Writing – original draft, Writing – review and editing; Helen Kurkjian, Conceptualization, Investigation, Visualization, Methodology, Writing – original draft, Writing – review and editing; Babak Momeni, Conceptualization, Software, Formal analysis, Supervision, Funding acquisition, Investigation, Visualization, Methodology, Writing – original draft, Project administration, Writing – review and editing

### Author ORCIDs
Helen Kurkjian http://orcid.org/0000-0002-3610-5344
Babak Momeni http://orcid.org/0000-0003-1271-5196

### Decision letter and Author response
Decision letter https://doi.org/10.7554/eLife.82504.sa1
Author response https://doi.org/10.7554/eLife.82504.sa2

## Additional files

### Supplementary files
• MDAR checklist

### Data availability
Codes used to generate the data in this study are shared on GitHub at https://github.com/bmomeni/spatial-coexistence (copy archived at *Momeni, 2022*).

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
