## [Editor Report]

This important study uses computational simulations to explore when spatial structure can promote the coexistence between different microbial species and when not, ultimately helping to explain diversity in microbial communities. The evidence supporting the conclusions is convincing, based on extensive parameter sweeps. The conclusion that spatial structure only promotes coexistence under certain conditions is a testable hypothesis that is very interesting to microbial ecologists quite broadly.

---

## [Decision Letter]

**Decision letter after peer review:**

Thank you for submitting your article "Spatial structure may favor or disfavor microbial coexistence" for consideration by *eLife*. Your article has been reviewed by 3 peer reviewers, and the evaluation has been overseen by a Reviewing Editor and Naama Barkai as the Senior Editor. The following individuals involved in the review of your submission have agreed to reveal their identity: Daniel R Amor (Reviewer #1).

Essential revisions:

As you can read in more detail below, the three reviewers all made similar recommendations to improve your manuscript. Please address at least the following points:

1) Justify the definition of coexistence (the 90% threshold);

2) Justify the way dispersal is implemented and the choice of diffusion coefficients;

3) Check for typos in the equations and the implementation of the carrying capacity;

4) Explore how relaxing these assumptions or using more realistic parameter values would affect current conclusions;

5) Explore not only coexistence but also population composition and the resulting spatial patterns.

*Reviewer #1 (Recommendations for the authors):*

I present here more concrete suggestions and concerns related to my comments in the public review.

1. I believe that there might be some errors in how the model's equations for species dynamics (line 298) were written in the manuscript. In the equation in line 299, should the term above the total cell carrying capacity (Ky) be a sum of all the species abundances instead of all the metabolites abundances? I believe that this is not the way that the matlab code was implemented, but it would be worth it for the authors to double-check this in all the scripts.

Ksat should have the same units as Cj(z,t), which is in disagreement with the units proposed in table 1. Similarly, Ky should have the same units as Si(z,t). Furthermore, it was not very clear how interactions, production, and consumption rates were assigned. Do they exist with a certain probability and, provided that they exist, their strength is sampled from a uniform distribution?

2. Line 321 defines the criteria to determine coexistence: 'only those with relative frequencies equal to or larger than 90% of the fastest growing species in the last 20 generations of the simulation are considered to coexist'. I am surprised by the relatively high (90%) threshold. This means that a pair of species that reaches a stable equilibrium of e.g. 10^7cells/ mL and 8*10^6 cells/mL is not classified as a pair of coexisting species. Please, discuss why the criteria imply such a high evenness of species abundances (experimental measures often report stable coexistence of species at many different fractions). I wonder if relaxing this strong requirement would significantly affect the results of community richness.

3. The text should be more clear from the beginning that the 'slow dispersal' scenario that is analyzed through the first sections (Figures1-3) is qualitatively very similar to a 'zero dispersal' scenario. Just to illustrate why I think this, one can use the Fisher's front speed solution v = 2*sqrt (r*D) to have an estimate (based on the proposed parameter values) of how much distance a monoculture could travel in the absence of interactions. The result, taking into account the timescale of the simulations (~350 hr, assuming r0 = 0.2/hr), is a total distance of ~0.02cm, which is consistent with the results in Figure S2. This means that, in this regime, microbial dynamics are mostly driven by growth, not by dispersal. On the other hand, considering the value of D in the 'fast dispersal' scenario gives an approximate front speed of 2*10^(-3)cm/hr (travelling ~0.7cm in 350 hours) for a monoculture in the absence of interactions. This is closer to observed speeds for colonies of non-motile bacteria growing on hard agar, and still falls on the slower end for such speeds. If one thinks about motile bacteria in soft agar, the speeds are even faster. Overall, I wonder about the kind of system that the 'slow dispersal' scenario could be modelling and, more importantly, whether the authors would consider analyzing the implications of 'fast dispersal' scenarios in more depth. How would Figures2 and 3 look under fast dispersal?

4. The figures of the main text could incorporate more analysis and results that further support the main message of each figure. Most of the figures contain only one panel (or one type of panel) and further information on the many parameters that could affect that result is left to the Supplementary Information. In some cases, just bringing some of the supplementary analysis to the main figures would make them more comprehensive while increasing the readability of the paper. Just to give some examples, figure 1 could incorporate the cartoon that explains the model (Figure S1), a time series, and some spatial profiles of species abundances to illustrate the typical dynamics of the system (e.g. incorporate some of the data in Figure S2). More snapshots of spatial profiles would help understand what happens in different scenarios, e.g. in Figure S6. In figures 2 and 3, how do these results depend on the average interaction strength (growth rate impact) of metabolites-to-bacteria?

5. The work will benefit from the additional analysis that can strengthen certain interpretations of the current results. For example, Figure S5 shows the dependence of species richness on the carrying capacity of the system. The authors claim that a shift towards intraspecies competition is responsible for such dependence and that a limited carrying capacity suppresses the more competitive species, but no statistical analysis on individual species performance is provided to back this argument (and species competitiveness is lacking a working definition). The criteria to determine the length of the simulations is another example, the authors said that 100 generations are 'often' enough to reach stability, but a more rigorous analysis or definition of stability would be better. Regarding the statement in line 116, an analysis of how much better or worse species grow when close to facilitative/inhibitory partners would be helpful.

6. The lack of analysis/visualization of spatial profiles for metabolite abundances leaves the reader wondering about the spatial scale of the interactions. Some representative cases could appear in the supplement, or be incorporated into one of the main figures.

*Reviewer #2 (Recommendations for the authors):*

The authors extend a mathematical model that they previously developed (Niehaus..Momeni, Nature Comm., 2019) for the study of microbial communities growing in well-mixed settings, to the case where species grow in spatial settings. They find that spatial structure promotes the coexistence of species when interactions are more facilitating than inhibiting, and when species dispersal is low. We found the paper well-written and well-organized. However, we have a number of comments: the main contribution of the paper is to extend a well-mixed model to a spatial model; It is thus fundamental that the assumptions made by the authors to model the spatial dynamics are well justified; we think that several physical parameters are chosen to values that do not represent realistic values for spatially structured communities and that the authors should discuss if the results hold also for more realistic values.

Line 84: The Authors state that they start each simulation from an initial distribution in which populations occupy adjacent, overlapping spatial locations at low initial density. What is the variation in the steady state distribution of species if they run many simulations with the same initial state and with the same parameters?

Line 88: "We have chosen 100 generations of growth because we have observed that often this is enough to reliably decide which species stably persist in the community." We think that the concept of generation is not clear. Can the authors define what generation means exactly? Also, can they provide a quantification for "reliably" when they say that the system converges reliably?

Line 91: The authors define a specific dilution procedure. We do not understand how they motivate this specific choice of dilution procedure. At each dilution step, they assume that the overall spatial distribution of the community is preserved and all populations at all locations are diluted by the same factor. Regarding this assumption, the authors say that they "adopt it as the least biased possibility, in the absence of additional information about a particular community." We have two questions regarding this assumption:

i) Is this really the least biased assumption? We suggest that a random distribution of the initial species is more a null model than the current choice. Or is there a reason to think that cells forming a new community by default inherit the spatial configuration of the parent community?

ii) Do the authors have a natural mechanism of community propagation in mind that they could refer to, which would correspond to their assumption?

Having in mind the concept of metacommunities, we don't understand how spatial distribution can be inherited by the new community.

Line 101: The authors say that a shift from competition to competition can favor coexistence in a spatially structured environment. When doing this analysis, the authors impose a cap on the total cell number that can exist at each location in space. The more restrictive the cap is, the more coexistence they find.

Regarding this conclusion, we would like the author to comment on the choice of the cap. The cap they pick is 10^9 cells/ml. This density is quite a low density for a spatially structured system. A back-of-the-envelope calculation can show that at 10^9 cells/ml, the volume ratio between cells and environment is 1:1000, if we consider that a cell occupies approximately 1 μm cube (e.g. *E. coli* in Minimal media+glucose is about that size. See: https://www.ncbi.nlm.nih.gov/books/NBK224751/). These are the order of magnitude estimates of course, but they suggest that the authors use a density that is much lower than expected in a spatially structured community, like a biofilm. In fact,10^9 cells/ml is about the density of cells in an overnight of *E. coli* grown in Minimal media (M9) with 0.2% glucose. We would suggest picking values closer to a dense community, and our expectation is that spatially structured communities should be at least 10^11 cells/ml. Following the calculation above, this would lead to a volume ratio between cells and the environment of 1:10, if we consider that a cell occupies 1 μm cube.

We would like the authors to comment on the following two points:

i) Why do they pick 10^9 cells/ml as a maximum cap, which seems such a low density?

ii) The authors pick 10^9 cells/ml because this cap maximizes coexistence. In light of the calculation we do here, what would happen if they used higher values of cell densities?

Line 101: 10^9 cells/ml is not 10^9 cells/cm as stated in Table 1. Can we think of these units as being the same? If yes, can units be homogenized? At the moment most of the units suggest that the effective dynamics is a 3D dynamics (e.g. diffusion constants, densities of cells), even if then they implement a 1D world, where there exists a line of "cubes" filled with sub-communities.

Table 1: The authors state that "Diffusion coefficient for mediators (DMed) 1.8 10-3 (cm2/hr)".

Again we do not understand the choice of parameters. When we look at realistic values of molecules diffusing, we see that these are more than 10 times faster. Here are two examples for glucose and for one amino acid:

For Glucose, the diffusion constant in water is 600 um^2 /sec, which is about 21*10-3 cm2/hr. See: https://bionumbers.hms.harvard.edu/bionumber.aspx?s=n&v=7&id=104089

For amino acids, the diffusion constant in water is 800 um^2/sec. See Wu, Y., Ma, P., Liu, Y. & Li, S. Diffusion coefficients of l-proline, l-threonine and l-arginine in aqueous solutions at 25◦C. Fluid Phase Equilibria 186, 27-38 (2001).

Table 1: Can they comment on the value of diffusion of species? This value represents somehow a fraction of individuals that move away from the local patch into another patch. What should we think this fraction to be?

Figure 1: what is the explanation for which, at a low fraction of facilitative interactions, coexistence in well-mixed and spatial communities is the same? Can they comment on this?

Figure 5: Do the authors expect that by increasing diffusion even further, coexistence should decrease again? Does the exploration stop at the maximum diffusion expected in liquid?

*Reviewer #3 (Recommendations for the authors):*

– I think there is a typo in the equations shown in the "Model description" section. In the population dynamics equation, the carrying capacity term (the one with k_Y) appears to contain a sum over the C_j rather than the expected sum over the S_i. In the supplied code, it appears that this term sums over the populations.

– Assuming the carrying capacity term is meant to contain a sum over the S_i rather than the C_j, the current formulation of this model may lead to non-physical outcomes. Consider a population that is above carrying capacity, such that the carrying capacity term is negative. If this population is also surrounded by highly detrimental chemical mediators (such that its interaction term is negative), the product of the negative carrying capacity and interaction terms will result in a positive growth rate. From the code I have looked at (Spatial1DInteraction_DpMM_ExMTC_flexibleTimeStep.m, lines 103-110), there doesn't seem to be any mechanism to prevent this. However, from the supplementary plots, it does not appear that the steady-state population abundances exceed the k_Y. The authors should assess whether these non-physical dynamics occur in their simulations, as it could artificially inflate coexistence. One possible solution is to set the population growth to zero if the carrying capacity is exceeded.

– One limitation of the authors' current analyses is that it is based only on richness, which does not reflect population abundance. Equivalent comparisons between the spatial and non-spatial models could be made with a metric like the Shannon entropy, which does consider population abundance. To compute the Shannon entropy in the spatial model, one could measure the total population of each cell type by integrating over the domain. From these abundances, a relative abundance distribution compatible with the Shannon entropy could be calculated. It would be worthwhile to assess whether the observed trends are similar when other metrics are used.

– I think it would be worthwhile for the authors to quantify the spatial patterns of coexistence in their model. I understand the authors' reasoning for focusing on a metric that does not consider spatial structure, as such metrics are the only ones that can be used to compare directly between spatial and well-mixed systems. However, from the plots shown in Figure S2, it appears that coexistence in this model can manifest as quite non-trivial spatial patterns. Spatial coexistence patterns of natural microbial communities are often strikingly beautiful, and it would be interesting to assess how this model's parameters influence its resulting spatial coexistence patterns. For example, one could examine the relationship between the mediator and microbe diffusion rates and the size/overlap of microbial domains, or analyze the existence of the seeming "dead-zones" of the domain seen in the bottom right panel of Figure S2.

– I find the result that changing the order of species in the initial condition can change the final richness to be very interesting (Figure S8). This result implies that there is a great deal of multistability in the dynamics. In a metapopulation context, this could be its own diversity-generating mechanism. I think discussing this multistability more explicitly in the main text would be worthwhile: is this multistability a result of spatial dynamics?

– The authors should specify the distribution of mediator production/consumption and interaction coefficients. Currently, I'm not sure what distribution these parameters are drawn from.

– In the figures with well-mixed vs. spatial heatmaps, it may be worthwhile to include a plot that directly depicts the ratio of the two model's richness values. As it stands, I found it a bit difficult to immediately see the differences between the well-mixed and spatial results.

---

## [Author Response]

Essential revisions:As you can read in more detail below, the three reviewers all made similar recommendations to improve your manuscript. Please address at least the following points:1) Justify the definition of coexistence (the 90% threshold);

As explained below in response to reviewers’ comments, in our definition of coexistence, we assess the changes in the relative frequency of species in the last 20 generations (compared to the fastest growing species) of our simulations. If the relative frequency of a species drops by more than 10% within 20 generations, we assume that they are destined to go extinct in the long-term. The idea is to eliminate species that are slowly going extinct, but are still present after 100 generations (the typical extent of each of our simulations). To be clear, there is no threshold on the steady state relative frequency of each species; i.e. the ratio of fastest growing to slowest growing coexisting species can be 1e6:1, 1:1, or 1:1e6, as long as the ratio does not change considerably during the last 20 generations of our simulations. We have reworded the sentence to 'only those species whose relative frequency do not drop by more than 10% compared to the fastest growing species in the last 20 generations of the simulation are considered to coexist.' We have also included Figure 4—figure supplement 1 as an example of a modified criterion to examine the final richness based on which species are present, rather than which species are stably present.

2) Justify the way dispersal is implemented and the choice of diffusion coefficients;

The primary reason for choosing the dispersal coefficients to be relatively small was to maintain the spatial structure of the community. This choice was made to highlight how spatial structure can contribute to coexistence. Our vision of an example of a community being discussed was a microbial mat in which there can be dispersal but a strong stratification is maintained. We agree with the reviewers that including a wider range of parameters would be informative; we have thus expanded our scope and included a wider range of parameters (including higher dispersal rates) in the revised version.

3) Check for typos in the equations and the implementation of the carrying capacity;

Thank you for pointing this out. The typo is fixed in the revised version.

4) Explore how relaxing these assumptions or using more realistic parameter values would affect current conclusions;

Our focus here is on general concepts, based on the relative rates of mediator diffusion and cell dispersal, rather than on attempting to recreate a particular community. Nevertheless, the choices made about the parameters are such that they are relevant to some examples of microbial communities such as microbial mats. We have included additional sentences to the introduction section to make it clear that in this paper we examine different ranges of parameters (some of them intentionally close to extremes) to explore different mechanisms and general principles.

5) Explore not only coexistence but also population composition and the resulting spatial patterns.

To address this comment, we have added more instances of spatial patterns of the communities along with coexistence results. We have also used Shannon index as a measure of evenness of the community in some of our comparisons of well-mixed versus spatial communities. We would like to emphasize that a detailed examination of the spatial organization of populations and metabolites within the community requires a dedicated investigation and is beyond the scope of this work. We plan to address these spatial organization aspects in future work.

Reviewer #1 (Recommendations for the authors):I present here more concrete suggestions and concerns related to my comments in the public review.1. I believe that there might be some errors in how the model's equations for species dynamics (line 298) were written in the manuscript. In the equation in line 299, should the term above the total cell carrying capacity (Ky) be a sum of all the species abundances instead of all the metabolites abundances? I believe that this is not the way that the matlab code was implemented, but it would be worth it for the authors to double-check this in all the scripts.Ksat should have the same units as Cj(z,t), which is in disagreement with the units proposed in table 1. Similarly, Ky should have the same units as Si(z,t). Furthermore, it was not very clear how interactions, production, and consumption rates were assigned. Do they exist with a certain probability and, provided that they exist, their strength is sampled from a uniform distribution?

Thank you for pointing out the typo. You are correct, that term is in fact the sum of all species abundances and we have confirmed that this was correctly implemented in our simulation codes in Matlab. We have fixed the Equation in the revised version.

We have added the following paragraph to clarify how the interactions and production/consumption rates were assigned (lines 393-402):

“To sample different possibilities, the interaction terms as well as production and consumption rates are randomly assigned in each instance of the simulation. Similar to our previous work [25], the production/consumption matrices are random, i.e. each element of the matrix has a binomial distribution with a fixed probability of being present (*q_p_* and *q_c_* for production and consumption/influence links, respectively). The production rate and consumption rates have a uniform distribution between 0.5 and 1.5 times a set value each (β_ij_ and α_ij_ for production and consumption rates, respectively). The interaction matrix which represents the influence of mediators on species has the same structure as the consumption matrix. The magnitude of the influence in this matrix has a uniform distribution between 0 and a maximum value, *r_int,0_*. The sign of the influence is chosen from a binomial distribution based on the ratio of fac:inh.”

2. Line 321 defines the criteria to determine coexistence: 'only those with relative frequencies equal to or larger than 90% of the fastest growing species in the last 20 generations of the simulation are considered to coexist'. I am surprised by the relatively high (90%) threshold. This means that a pair of species that reaches a stable equilibrium of e.g. 10^7cells/ mL and 8*10^6 cells/mL is not classified as a pair of coexisting species. Please, discuss why the criteria imply such a high evenness of species abundances (experimental measures often report stable coexistence of species at many different fractions). I wonder if relaxing this strong requirement would significantly affect the results of community richness.

Sorry for the confusion. The 90% threshold applies to the changes in the relative frequency in the last 20 generations compared to the fastest growing species. Coexisting species can have any relative frequency compared to each other (1:1 or 1:10^6^) in terms of the composition of the community. However, if the relative frequency of a population drops by more than 10% in 20 generations, we assume that they are destined to go extinct in the long-term. The idea is to eliminate species that are slowly going extinct, but are still present after 100 generations which is the typical extent of each run. We have reworded the sentence to the following (lines 412-420):

“To assess coexistence, we use a criterion similar to [25]. In short, any species whose density drops below a pre-specified extinction threshold (ExtTh) is considered extinct. Among species that persist throughout the simulation, only those with relative frequencies equal to or larger than 90% of the fastest growing species in the last 20 generations of the simulation are considered to coexist. We consider these species to be ‘stably present’ in the community. Species with a declining relative frequency are assumed to go extinct later and are not considered to be part of coexisting communities. The only exception to this criterion is the data in Figure 4—figure supplement 1, in which all ‘present’ species (rather than stably present species) are included in the assessment of final richness.”

We have also changed the terminology to call species that satisfy this criterion and coexist in a community as ‘stably present’ to make the distinction clearer.

3. The text should be more clear from the beginning that the 'slow dispersal' scenario that is analyzed through the first sections (Figures1-3) is qualitatively very similar to a 'zero dispersal' scenario. Just to illustrate why I think this, one can use the Fisher's front speed solution v = 2*sqrt (r*D) to have an estimate (based on the proposed parameter values) of how much distance a monoculture could travel in the absence of interactions. The result, taking into account the timescale of the simulations (~350 hr, assuming r0 = 0.2/hr), is a total distance of ~0.02cm, which is consistent with the results in Figure S2. This means that, in this regime, microbial dynamics are mostly driven by growth, not by dispersal. On the other hand, considering the value of D in the 'fast dispersal' scenario gives an approximate front speed of 2*10^(-3)cm/hr (travelling ~0.7cm in 350 hours) for a monoculture in the absence of interactions. This is closer to observed speeds for colonies of non-motile bacteria growing on hard agar, and still falls on the slower end for such speeds. If one thinks about motile bacteria in soft agar, the speeds are even faster. Overall, I wonder about the kind of system that the 'slow dispersal' scenario could be modelling and, more importantly, whether the authors would consider analyzing the implications of 'fast dispersal' scenarios in more depth. How would Figures2 and 3 look under fast dispersal?

We agree with the reviewer about the range of dispersal represented in Figures1-3. The low dispersal regime was initially chosen as a way to make a clear distinction from the well-mixed condition and exaggerate the spatial aspects. We would like to emphasize that even changes in this regime, which would be perhaps considered minor differences in motility, still have an impact on coexistence. We have included additional results of ‘fast dispersal’ in the supplementary information of the revised manuscript (Figure 1—figure supplement 2). We have also expanded Figure 4 to include ‘fast dispersal’ and added the corresponding discussions and two supplementary figures (Figure 4—figure supplement 1 and Figure 4—figure supplement 2) in the revised manuscript (lines 236-274).

“At intermediate levels of dispersal, the trend reversed and well-mixed communities showed more coexistence compared to spatial communities. This is interesting because at the limit of extremely rapid diffusion (shown with a ‘∞’ sign in Figure 4) when we kept the species distribution uniform across the spatial extent, coexistence outcomes matched the well-mixed case, as expected. We found two factors that contributed to this trend. The first contribution came from longer-term changes in dynamics at intermediate levels of dispersal. Even after 100 generations, which is the typical extent of our studies, at intermediate levels of dispersal (e.g. DCell = 5×10-6 cm2/hr), the spatial distribution of populations is still changing considerably. As a result, our strict criteria for stable coexistence removes some of the populations that are still temporally not stable enough, leading to a lower overall assessment of coexistence in these cases. To show this, we examined the range of dispersal coefficients again, but kept all the species that were present after 100 generations, rather than those with stable population fractions at that point (see Model implementation in Methods). The results show that higher dispersal coefficients using this measure lowers the richness of resulting communities (based on presence, rather than stable presence), but not below the levels expected from well-mixed communities (Figure 4—figure supplement 1). As a second factor, we hypothesized that self-facilitation interactions contribute to the decrease in coexistence at intermediate dispersal levels. Our rationale was that self-facilitation interactions are amplified in communities in which the spatial context is preserved, because the distribution of producers matches the distribution of self, but not other recipients in such a case. This can lead to community overtake by a self-facilitating species. This effect will be weaker in communities at intermediate dispersal rates: at low dispersal rates self-facilitating species will be more confined in space and some of the metabolite will leak out to other species; in the other extreme, in the very high dispersal rates all distributions will become uniform and the distinction between self and others diminishes. To test this, we tested weaker self-facilitation links in our simulations and observed that this change led to higher coexistence in communities with intermediate dispersal coefficients but not in well-mixed communities or communities with low dispersal coefficients (Figure 4—figure supplement 2). It is a matter of debate how prevalent self-facilitation interactions are within microbial communities. Self-facilitation interactions do exist, for example when a species breaks down a recalcitrant substrate such as cellulose into smaller molecules that can be beneficial. However, if they are not as prevalent as what our model assumes, some of our predictions might be affected.”

4. The figures of the main text could incorporate more analysis and results that further support the main message of each figure. Most of the figures contain only one panel (or one type of panel) and further information on the many parameters that could affect that result is left to the Supplementary Information. In some cases, just bringing some of the supplementary analysis to the main figures would make them more comprehensive while increasing the readability of the paper. Just to give some examples, figure 1 could incorporate the cartoon that explains the model (Figure S1), a time series, and some spatial profiles of species abundances to illustrate the typical dynamics of the system (e.g. incorporate some of the data in Figure S2). More snapshots of spatial profiles would help understand what happens in different scenarios, e.g. in Figure S6. In figures 2 and 3, how do these results depend on the average interaction strength (growth rate impact) of metabolites-to-bacteria?

Thank you for the suggestion. We have combined former Figure S1 with Figure 1, in response to your suggestion. To make the relevant figures more accessible, we have also renamed the supplementary figures as Figure 1-S1, Figure 1-S2, etc. to take advantage of the *eLife* format that aggregates all supplementary figures related to each main figure. We believe this format will make it easier for the reader to see additional aspects related to the message of each figure (which are provided in supplementary figures).

We have also added supplementary figures to show that the trends in Figures2 and 3 are preserved at weaker and stronger interaction levels (Figure 2—figure supplement 2, Figure 2—figure supplement 3, Figure 3figure supplement 2, and Figure 3—figure supplement 3).

Figure 1—figure supplement 3 in the revised version addresses your concern about more snapshots of the spatial profile by comparing the low-dispersal and high-dispersal cases.

5. The work will benefit from the additional analysis that can strengthen certain interpretations of the current results. For example, Figure S5 shows the dependence of species richness on the carrying capacity of the system. The authors claim that a shift towards intraspecies competition is responsible for such dependence and that a limited carrying capacity suppresses the more competitive species, but no statistical analysis on individual species performance is provided to back this argument (and species competitiveness is lacking a working definition). The criteria to determine the length of the simulations is another example, the authors said that 100 generations are 'often' enough to reach stability, but a more rigorous analysis or definition of stability would be better. Regarding the statement in line 116, an analysis of how much better or worse species grow when close to facilitative/inhibitory partners would be helpful.

To show our rationale behind the choice of 100 generations as a representative time for the outcome of the communities, we have added a supplementary figure (Figure 1—figure supplement 1). It demonstrates that the community composition after 100 generations typically only undergoes small fluctuations in both well-mixed and spatial communities.

Regarding the statement in line 116, we have added Figure 1—figure supplement 6 as an example of commensalism when the distance between the provider and recipient was changed. In this example, the benefit received depended on the diffusion of the mediator. As a result, stronger interaction was observed when species are closer to each other.

6. The lack of analysis/visualization of spatial profiles for metabolite abundances leaves the reader wondering about the spatial scale of the interactions. Some representative cases could appear in the supplement, or be incorporated into one of the main figures.

Thank you for the suggestion. We have added more representative cases of both spatial patterns (Figure 1—figure supplement 3) and community dynamics (Figure 1—figure supplement 4 and Figure 1—figure supplement 5) in the revised version of the manuscript.

Reviewer #2 (Recommendations for the authors):The authors extend a mathematical model that they previously developed (Niehaus..Momeni, Nature Comm., 2019) for the study of microbial communities growing in well-mixed settings, to the case where species grow in spatial settings. They find that spatial structure promotes the coexistence of species when interactions are more facilitating than inhibiting, and when species dispersal is low. We found the paper well-written and well-organized. However, we have a number of comments: the main contribution of the paper is to extend a well-mixed model to a spatial model; It is thus fundamental that the assumptions made by the authors to model the spatial dynamics are well justified; we think that several physical parameters are chosen to values that do not represent realistic values for spatially structured communities and that the authors should discuss if the results hold also for more realistic values.

Thank you for your constructive feedback. We have revised the manuscript to explain the model assumptions and fixed some of the errors that had created the impression that the model parameters were not realistic.

Line 84: The Authors state that they start each simulation from an initial distribution in which populations occupy adjacent, overlapping spatial locations at low initial density. What is the variation in the steady state distribution of species if they run many simulations with the same initial state and with the same parameters?

With the same initial state and the same parameters, the simulations are deterministic and there is no stochasticity in the results. The stochasticity is intentionally introduced in the properties of the species (e.g. basal growth rates, rates of consumption and production, etc.) to assess the outcomes in many instances of simulated communities.

Line 88: "We have chosen 100 generations of growth because we have observed that often this is enough to reliably decide which species stably persist in the community." We think that the concept of generation is not clear. Can the authors define what generation means exactly? Also, can they provide a quantification for "reliably" when they say that the system converges reliably?

Generation is defined as the amount of time in which the sum of population density of all species doubles. Practically, the number of generations in each time step (dt) is calculated as ln(ΣS_t+dt_/ΣS_t_)/ln(2). We use this convention to describe the progression of communities in a unified way, instead of using the actual time which will need to be normalized between cases with different growth rates.

To show our rationale behind the choice of 100 generations as a representative time for the outcome of simulated communities, we have added a supplementary figure (Figure 1—figure supplement 1). It demonstrates that the community composition after 100 generations typically only undergoes small fluctuations in both well-mixed and spatial communities.

Line 91: The authors define a specific dilution procedure. We do not understand how they motivate this specific choice of dilution procedure. At each dilution step, they assume that the overall spatial distribution of the community is preserved and all populations at all locations are diluted by the same factor. Regarding this assumption, the authors say that they "adopt it as the least biased possibility, in the absence of additional information about a particular community." We have two questions regarding this assumption:i) Is this really the least biased assumption? We suggest that a random distribution of the initial species is more a null model than the current choice. Or is there a reason to think that cells forming a new community by default inherit the spatial configuration of the parent community?ii) Do the authors have a natural mechanism of community propagation in mind that they could refer to, which would correspond to their assumption?Having in mind the concept of metacommunities, we don't understand how spatial distribution can be inherited by the new community.

We acknowledge that this choice is one among several possible options. One motivation for this choice was to preserve the spatial patterns that set spatial communities apart from well-mixed communities. We have revised the text to justify our choice and address your concern (lines 95102):

“At each dilution step, we assume that the overall spatial distribution of the community is preserved and all populations at all locations are diluted with the same factor. We recognize that this assumption is not universally true; however, we adopt it as an approximation, in the absence of additional information about a particular community. Such a dilution preserves some of the spatial structure of the community in the next round of growth and could represent a biofilm getting partially washed away by rain or in a microfluidic device, gut microbiota after a defecation event, or a broken-off portion of a granule initiating a new granule.”

Line 101: The authors say that a shift from competition to competition can favor coexistence in a spatially structured environment. When doing this analysis, the authors impose a cap on the total cell number that can exist at each location in space. The more restrictive the cap is, the more coexistence they find.Regarding this conclusion, we would like the author to comment on the choice of the cap. The cap they pick is 10^9 cells/ml. This density is quite a low density for a spatially structured system. A back-of-the-envelope calculation can show that at 10^9 cells/ml, the volume ratio between cells and environment is 1:1000, if we consider that a cell occupies approximately 1 μm cube (e.g. *E. coli* in Minimal media+glucose is about that size. See: https://www.ncbi.nlm.nih.gov/books/NBK224751/). These are the order of magnitude estimates of course, but they suggest that the authors use a density that is much lower than expected in a spatially structured community, like a biofilm. In fact,10^9 cells/ml is about the density of cells in an overnight of *E. coli* grown in Minimal media (M9) with 0.2% glucose. We would suggest picking values closer to a dense community, and our expectation is that spatially structured communities should be at least 10^11 cells/ml. Following the calculation above, this would lead to a volume ratio between cells and the environment of 1:10, if we consider that a cell occupies 1 μm cube.We would like the authors to comment on the following two points:i) Why do they pick 10^9 cells/ml as a maximum cap, which seems such a low density?ii) The authors pick 10^9 cells/ml because this cap maximizes coexistence. In light of the calculation we do here, what would happen if they used higher values of cell densities?

The focus of our investigation was on the trend, rather than the exact value of the cap. There is an aspect of arbitrariness in our choice of the cap. Similarly, while we agree with the reviewer’s calculations, the choice of *E. coli* for the calculations is arbitrary. These numbers will be different by around two orders of magnitude, for example if we consider a larger yeast cell (~4 μm cube) as the unit. The value of the cap only becomes important in relation to the other parameters of choice. In our simulations we have chosen k_Y_ to be large enough to not affect the coexistence outcomes. Any value of k_Y_ above our chosen value will have little impact on the outcome. Instead, lower values of k_Y_ will have an impact. k_Y_ becomes important if some of the species reach this cap. Under such a condition, the fastest growing species will slow down when it reaches the cap, allowing other species to catch up before the next dilution step. As a result, more coexistence will be supported when k_Y_ is more restrictive, as shown in Figure 1—figure supplement 8.

Line 101: 10^9 cells/ml is not 10^9 cells/cm as stated in Table 1. Can we think of these units as being the same? If yes, can units be homogenized? At the moment most of the units suggest that the effective dynamics is a 3D dynamics (e.g. diffusion constants, densities of cells), even if then they implement a 1D world, where there exists a line of "cubes" filled with sub-communities.

Your interpretation is correct. To allow direct comparison between the spatial and well-mixed cases, we have formulated all equations based on the three-dimensional (3D) densities of cells and concentrations of mediators. All the densities in both spatial and well-mixed simulations are based on volumetric densities of cells, matching your description as a line of cubes (with variations only in a single dimension).

Table 1: The authors state that "Diffusion coefficient for mediators (DMed) 1.8 10-3 (cm2/hr)".Again we do not understand the choice of parameters. When we look at realistic values of molecules diffusing, we see that these are more than 10 times faster. Here are two examples for glucose and for one amino acid:For Glucose, the diffusion constant in water is 600 um^2 /sec, which is about 21*10-3 cm2/hr. See: https://bionumbers.hms.harvard.edu/bionumber.aspx?s=n&v=7&id=104089For amino acids, the diffusion constant in water is 800 um^2/sec. See Wu, Y., Ma, P., Liu, Y. & Li, S. Diffusion coefficients of l-proline, l-threonine and l-arginine in aqueous solutions at 25◦C. Fluid Phase Equilibria 186, 27-38 (2001).

Our apologies for the typo. The typical diffusion coefficient for mediators (D_Med_) is 1.8 x 10^-2^ cm^2^/hr (or 500 µm^2^ /sec), which is consistent with the reports you have listed. The correct value was used in the simulations, but the wrong value was listed in Table 1. We have fixed this typo in the revised version.

Table 1: Can they comment on the value of diffusion of species? This value represents somehow a fraction of individuals that move away from the local patch into another patch. What should we think this fraction to be?

The dispersal coefficient (D_Cell_) as defined in our manuscript represents a one-dimensional random walk. In the overly simplified case that the population cell density is a spike of with no neighboring cells present, the fraction of cells transferred to a small distance Del_z in each direction within a small time-step Δt is D_Cell_*Δt/(Δz)^2. If we set the parameters to D_Cell_ = 5e-9 cm^2^/hr, Δt = 0.01 hr, and Δz = 0.005 cm, this fraction is 2e-6. Numerically, this fraction has to be small, so that we can track the dispersal process accurately.

Figure 1: what is the explanation for which, at a low fraction of facilitative interactions, coexistence in well-mixed and spatial communities is the same? Can they comment on this?

Between the two mechanisms of self-organization, co-localization driven by facilitation versus segregation driven by inhibition, the former has a stronger effect on coexistence in our simulations. The positive influence is further enforced by more growth in the vicinity of the partner, leading to a stronger representation of facilitation in spatial communities. In contrast, segregation only has a modest effect on weakening the impact of inhibition. As a result, there is more similarity between well-mixed and spatial communities in the absence of strong facilitative interactions. We have added this explanation to the revised text.

Figure 5: Do the authors expect that by increasing diffusion even further, coexistence should decrease again? Does the exploration stop at the maximum diffusion expected in liquid?

We have updated Figure 5 with higher values of the diffusion coefficient. Even at diffusion coefficients faster than what we would expect to see in practice, coexistence remained steady. We agree with the reviewer that if diffusion is fast enough the results should resemble the well-mixed situation. However, with the parameters selected for our configuration and with physically plausible diffusion coefficients we do not reach that regime.

Reviewer #3 (Recommendations for the authors):– I think there is a typo in the equations shown in the "Model description" section. In the population dynamics equation, the carrying capacity term (the one with k_Y) appears to contain a sum over the C_j rather than the expected sum over the S_i. In the supplied code, it appears that this term sums over the populations.

Thank you for bringing this to our attention. We have fixed the typo in that equation.

– Assuming the carrying capacity term is meant to contain a sum over the S_i rather than the C_j, the current formulation of this model may lead to non-physical outcomes. Consider a population that is above carrying capacity, such that the carrying capacity term is negative. If this population is also surrounded by highly detrimental chemical mediators (such that its interaction term is negative), the product of the negative carrying capacity and interaction terms will result in a positive growth rate. From the code I have looked at (Spatial1DInteraction_DpMM_ExMTC_flexibleTimeStep.m, lines 103-110), there doesn't seem to be any mechanism to prevent this. However, from the supplementary plots, it does not appear that the steady-state population abundances exceed the k_Y. The authors should assess whether these non-physical dynamics occur in their simulations, as it could artificially inflate coexistence. One possible solution is to set the population growth to zero if the carrying capacity is exceeded.

Thank you for pointing this out. Because in our typical runs we had simulated the populations growing to saturation from an initially low density, we never encountered the non-physical situation that the reviewer has correctly pointed out. To avoid potential issues in the future, we have revised the code. In the revised code, the baseline growth rate is used instead of the net growth rate of the interactions whenever a population density exceeds its corresponding carrying capacity. The updated codes are posted to GitHub for future reference.

– One limitation of the authors' current analyses is that it is based only on richness, which does not reflect population abundance. Equivalent comparisons between the spatial and non-spatial models could be made with a metric like the Shannon entropy, which does consider population abundance. To compute the Shannon entropy in the spatial model, one could measure the total population of each cell type by integrating over the domain. From these abundances, a relative abundance distribution compatible with the Shannon entropy could be calculated. It would be worthwhile to assess whether the observed trends are similar when other metrics are used.

We have repeated our analysis using Shannon entropy as the measure of diversity. We have reported the results in a supplementary figure (Figure 1—figure supplement 2 in the revised version), showing that a trend similar to mean richness is observed, but the magnitude of the effect is smaller with the Shannon index.

**Author response image 1. sa2fig1:** 

– I think it would be worthwhile for the authors to quantify the spatial patterns of coexistence in their model. I understand the authors' reasoning for focusing on a metric that does not consider spatial structure, as such metrics are the only ones that can be used to compare directly between spatial and well-mixed systems. However, from the plots shown in Figure S2, it appears that coexistence in this model can manifest as quite non-trivial spatial patterns. Spatial coexistence patterns of natural microbial communities are often strikingly beautiful, and it would be interesting to assess how this model's parameters influence its resulting spatial coexistence patterns. For example, one could examine the relationship between the mediator and microbe diffusion rates and the size/overlap of microbial domains, or analyze the existence of the seeming "dead-zones" of the domain seen in the bottom right panel of Figure S2.

Quantifying and investigating the spatial patterns and their link to creation/depletion of spatial niches by other species is certainly interesting and worthwhile. However, it requires extensive work that we feel distracts from the current message of this manuscript. Our lab is pursuing this line of study, but we feel a thorough investigation of spatial patterns is beyond the scope of this paper.

– I find the result that changing the order of species in the initial condition can change the final richness to be very interesting (Figure S8). This result implies that there is a great deal of multistability in the dynamics. In a metapopulation context, this could be its own diversity-generating mechanism. I think discussing this multistability more explicitly in the main text would be worthwhile: is this multistability a result of spatial dynamics?

We agree with the reviewer that the observation of multiple distinct stable states in these spatial communities is tantalizing. However, since this paper focused on the comparison of well-mixed versus spatial communities, detailed discussions of the spatial distributions and their impact on coexistence is beyond the scope of this manuscript. We plan to address these aspects in follow-up work.

– The authors should specify the distribution of mediator production/consumption and interaction coefficients. Currently, I'm not sure what distribution these parameters are drawn from.

This point was also a concern for Reviewer #1. We have added a description to clarify this point in the revised manuscript (lines 393-402):

“To sample different possibilities, the interaction terms as well as production and consumption rates are randomly assigned in each instance of the simulation. Similar to our previous work [25], the production/consumption matrices are random, i.e. each element of the matrix has a binomial distribution with a fixed probability of being present (*q_p_* and *q_c_* for production and consumption/influence links, respectively). The production rate and consumption rates have a uniform distribution between 0.5 and 1.5 times a set value each (β_ij_ and α_ij_ for production and consumption rates, respectively). The interaction matrix which represents the influence of mediators on species has the same structure as the consumption matrix. The magnitude of the influence in this matrix has a uniform distribution between 0 and a maximum value, *r_int,0_*. The sign of the influence is chosen from a binomial distribution based on the ratio of fac:inh.”

– In the figures with well-mixed vs. spatial heatmaps, it may be worthwhile to include a plot that directly depicts the ratio of the two model's richness values. As it stands, I found it a bit difficult to immediately see the differences between the well-mixed and spatial results.

Thank you for the suggestion. In the revised version we have included the ratios of richness values as additional figures associated with Figures 2 and 3.